# Impact of Exercise-Induced Pulmonary Hypertension on Right Ventricular Function and on Worsening of Cardiovascular Risk in HIV Patients

**DOI:** 10.3390/jcm11247349

**Published:** 2022-12-10

**Authors:** Rosalinda Madonna, Lorenzo Ridolfi, Riccardo Morganti, Filippo Biondi, Silvia Fabiani, Arianna Forniti, Riccardo Iapoce, Raffaele De Caterina

**Affiliations:** 1Institute of Cardiology, Department of Pathology, Cardiology Division, Azienda Ospedaliera Universitaria Pisana, University of Pisa, 56124 Pisa, Italy; 2Institute of Epidemiology, University of Pisa, 56124 Pisa, Italy; 3Infectious Disease Unit, Department of Clinical and Experimental Medicine, Azienda Ospedaliera Universitaria Pisana, University of Pisa, 56124 Pisa, Italy

**Keywords:** exercise-induced pulmonary hypertension, acquired immunodeficiency syndrome, cardio-pulmonary exercise test, echocardiography, cardiovascular risk

## Abstract

Background and Aim: Exercise-induced pulmonary hypertension (ExPH) predicts clinical outcomes, such as all-cause mortality and cardiovascular (CV) hospitalizations, in patients with dyspnea on effort. We investigated its prognostic significance in human immunodeficiency virus (HIV)-affected patients. Methods: In 52 consecutive HIV patients with either low (n = 47) or intermediate probability (n = 5) of PH at rest, we evaluated—at time 0 and after 2 years—the prognostic determinants of CV risk, according to the 2015 European Society of Cardiology (ESC)/European Respiratory Society (ERS) Guidelines. Patients were classified with or without ExPH at stress echocardiography (ESE) and cardiopulmonary exercise test (CPET). We then related ExPH at time 0 with clinical worsening (CV risk score increase >20% after 2 years). Results: Right ventricle (RV) systolic function was significantly reduced in patients with ExPH compared to those without ExPH at CPET. This also occurred in patients with intermediate/high probability compared to those with low probability of ExPH at ESE. The former exhibited worse values of TAPSE and FAC (*p* < 0.001 and *p* = 0.01, respectively). A significantly higher proportion of patients with ExPH (CPET) or with intermediate/high probability of ExPH (ESE) had higher sPAP (*p* < 0.001), mPAP (*p* = 0.004) and higher TRV (*p* = 0.006), as well as higher right atrial area (*p* < 0.001) and indexed right atrial volume (*p* = 0.004). Total pulmonary vascular resistance (expressed by the ratio between TRV and the velocity-time integral at the level of the right ventricular outflow tract) was higher both in patients with ExPH and in those with intermediate/high probability of ExPH (*p* < 0.001). Patients with intermediate/high probability of ExPH at ESE showed a trend (*p* = 0.137) towards clinical worsening compared to those with low probability of ExPH. No patients with low probability of ExPH had a >20% increased CV risk score after 2 years. We found an association between higher NT-proBNP and the presence or intermediate/high probability of ExPH after 2 years (*p* = 0.048 at CPET, *p* = 0.033 at ESE). Conclusions: The assessment of ExPH may predict a trend of increasing CV risk score over time. If confirmed at a longer follow-up, ExPH could contribute to better risk stratification in HIV patients.

## 1. Introduction

Pulmonary hypertension (PH), particularly the pre-capillary hemodynamic form, may complicate multiple clinical conditions that include systemic, acquired and infectious diseases, such as Human Immunodeficiency Virus (HIV) infection. Pre-capillary PH is currently defined as an increase in mean pulmonary artery pressure (mPAP) >20 mmHg, pulmonary artery wedge pressure (PAWP) <15 mmHg and a value of pulmonary vascular resistances >2 Woods units as assessed by right heart catheterization (RHC) at rest [1]. HIV infection is a risk factor for pre-capillary PAH [1], which in turn is a cause of increased mortality among patients with HIV [2]. In fact, the remarkable increase in early diagnosis, the introduction of highly active antiretroviral therapy (ART) and the aggressive management of opportunistic infections allowed increased life expectancy in HIV patients. As a result, chronic diseases, including cardiovascular disease and PH, are the major cause of mortality and morbidity in these patients, in which the prevalence of PH reached 0.46% [3,4]. The exact mechanisms by which PAH develops in subjects with HIV are not fully understood [1]. However, previous studies suggest that the HIV role in the pathogenesis of PAH is not related to viral load or direct infection of endothelial cells, but rather to the immune dysregulation promoted by HIV infection, including both impaired cellular immunity and a chronic inflammatory condition, as well as to individual susceptibility to PAH [1,5].

One of the previous definitions of “exercise-induced PH” (exPH) is the increase of systolic pulmonary artery pressure (sPAP) by at least 20 mmHg during a low-intensity exercise (e.g., 50 W) at exercise stress echocardiogram (ESE) or cardiopulmonary stress test (CPET) [6]. Current evidence shows that exercise-induced pulmonary hypertension (ExPH) may predict clinical outcomes, such as all-cause mortality or cardiovascular hospitalizations, in patients with dyspnea on effort [7]. In clinical groups who are at risk for PH—for example, patients with systemic sclerosis and patients with left heart disease **[8,9,10,11]**—ExPH may indicate an early stage of pulmonary vascular disease and, therefore, correlate with reduced survival, similar to individuals with resting PAH. We have recently demonstrated that ExPH is associated with a worse clinical status, a longer time to HIV diagnosis and to beginning of ART, and a poorer immunological control of HIV infection, evaluated by CD4+ T cell count [12]. In the same group of subjects, ExPH is associated with a high cardiovascular (CV) risk score [13], as assessed by the presence/absence of the prognostic determinants of CV risk, according to the 2015 European Society of Cardiology (ESC)/European Respiratory Society (ERS) Guidelines [14]. This suggested a potential role of ExPH in assessing the CV risk of HIV patients. Here, an integrated diagnostic work-out that included an exercise stress echocardiogram (ESE) and cardiopulmonary stress test (CPET) allowed us to recognize early HIV patients who develop ExPH, the presence of which is associated with a worse World Health Organization functional class (WHO-FC), a shorter walk distance at 6-min walking test (6MWT) and a lower VO_2_ peak [10]. Finally, we found that those individuals who had a worse immunological control of the disease also had a worse CV risk profile [10], confirming the role of a weakened or dysfunctional immune system in determining the prognosis and outcomes of HIV patients [2,3,10,15].

In the present study, we hypothesized that ExPH might be associated with a worsening of CV risk score over time in HIV patients and may be useful for risk stratification in the follow-up. We aimed to assess whether the presence of ExPH correlates with a worse prognosis at 2 years, measured as worsening of the CV risk score >20% compared to the baseline. Therefore, HIV patients with ExPH, assessed non-invasively by ESE and CPET, and without a high probability of PH at echocardiogram at rest, underwent a 2-year follow-up that included a reassessment of the CV risk score.

## 2. Methods

### 2.1. Study Design and Data Collection

This was a prospective, observational, cohort study, enrolling 52 patients with HIV recruited from the Infectious Disease Unit, Department of Clinical and Experimental Medicine, Azienda Ospedaliera Universitaria Pisana, University of Pisa, Italy. The study complied with the Helsinki Declaration, and informed consent was obtained from all patients before each diagnostic test, done for clinical purposes. Local investigators had full access to patient data and medical records.

All enrolled patients underwent transthoracic echocardiography (TTE) at rest and stress echocardiography (ESE) prior to a cardiopulmonary exercise test (CPET). We included n = 47 patients with “low” PH probability at rest and n = 5 patients with “intermediate” PH probability at rest. Eight patients with “high” PH probability at rest were excluded. Additional exclusion criteria were the presence of moderate-to-severe anemia (hemoglobin < 10 g/dL); significant left heart disease at resting echocardiogram; history of venous thromboembolism, kidney disease, or chronic obstructive pulmonary disease; or moderate to severe tricuspid insufficiency.

All patients included in the study underwent CPET and ESE and were classified as either with or without ExPH. We evaluated an overall CV risk score at time 0 and after 2 years in all patients, as previously reported [13] and as detailed in the online supplement. We then evaluated the association of a higher CV score at baseline and after two years follow-up with the presence/absence of ExPH and with several echocardiographic parameters.

### 2.2. Echocardiography at Rest

According to the recently updated ESC/ERS PH guidelines, we assessed the peak of tricuspid regurgitation velocity (TRV) to determine the echocardiographic probability of PH, and eventually additional echocardiographic PH signs, as previously described [5,10,16,17,18] and detailed in the Appendix A. Patients were thus classified as either at low, intermediate or high probability of PH at rest.

### 2.3. Stress Echocardiography

Stress echocardiography (ESE) was performed according to the protocol recommended by the European Association of Echocardiography (EAE) guidelines [16] and as detailed in the Appendix A. Patients were thus classified as either at low, intermediate or high probability of ExPH. Groups were compared at baseline and after two years follow-up.

### 2.4. Cardiopulmonary Stress Test

We performed a symptom-limited CPET on an electronically-braked cycle ergometer using a ramp-pattern increase in work rate, as previously described [5,10,16,17,18] and detailed in the Appendix A. The two groups of patients were compared at baseline and after two years follow-up.

### 2.5. Statistical Analysis

Categorical and continuous data were described by absolute frequency and mean ± standard deviation (SD), respectively. Qualitative and quantitative parameters were compared by the chi square test and t-test for independent samples (two-tailed), respectively, with presence or absence of ExPH as diagnosed at ESE and CPET. “Clinical worsening” was assessed based on the presence or absence of an increase in the CV risk score >20% after 2 years, then compared with qualitative and quantitative parameters by Fisher’s exact test and the *t*-test for independent samples (two-tailed), respectively. The statistical significance level was set at 0.05. All analyses were performed by SPSS v.28 software, Chicago, IL, USA.

## 3. Results

Between 1 November 2019, and 31 August 2022, 52 patients with HIV were recruited from the Infectious Disease Unit, Department of Clinical and Experimental Medicine, Azienda Ospedaliera Universitaria Pisana, University of Pisa, Italy. Six patients were excluded at follow-up due to high probability of PH. Out of the 46 patients included, 12 and 24 showed ExPH at baseline CPET and ESE, respectively, while the remaining subjects were negative at both exercise evaluations. At baseline, there were no significant differences between the two groups in terms of age, body surface area, heart rate, history of syncope, systolic blood pressure, history of high blood pressure and concomitant medications, such as *beta-blockers, aspirin, statins, oral anticoagulants or digoxin* (Appendix A). None of the patients received any specific PAH treatment.

In Table 1 and Table 2 we report the echocardiographic parameters at 2 years, measured by resting echocardiography 10 min after the execution of the cardiopulmonary test. Echocardiography did not show dilatation of the right ventricle (RV) both in patients with and without ExPH at CPET (Table 1), and there were no significant differences between the two groups in terms of left ventricular (LV) function assessed by LV ejection fraction (*p* = 0.908). However, RV systolic function was significantly reduced in patients with ExPH, who exhibited worse values of TAPSE and FAC (*p* < 0.001 and *p* = 0.01, respectively) (Table 1). A significantly higher proportion of patients with ExPH had higher sPAP (*p* < 0.001), mPAP (*p* = 0.004) and higher TRV (*p* = 0.006), as well as higher right atrial area (*p* < 0.001) and indexed right atrial volume (*p* = 0.004). Furthermore, total pulmonary vascular resistance (expressed by the ratio between TRV and the velocity-time integral at the level of the right ventricular outflow tract) was higher in patients with ExPH (*p* < 0.001) (Table 1). Comparing the echocardiographic parameters of patients with low probability of ExPH at ESE with those having intermediate or high probability of ExPH, we observed the same differences, except for RV systolic function, as TAPSE and FAC were non-significantly inferior in patients with intermediate or high probability of ExPH compared with low probability of ExPH (*p* = 0.186 and *p* = 0.092, respectively) (Table 2).

The average duration of follow-up was 2 years. During this time, no patients died. All 45 patients in the study underwent a CV risk score evaluation at baseline and after 2 years. At the end of the follow-up, we assessed the presence of a “clinical worsening” as it was previously defined [13]. Stratified on the PH probability during exercise at ESE, patients had a different trend of progression on CV risk score at 2 years, with clinical worsening occurring more frequently in patients with intermediate/high probability of ExPH than in those with low probability (Figure 1). At baseline, the former group showed an average score of 11 ± 2, while the latter was 9 ± 2. At the end of the follow-up, individuals with intermediate/high PH probability showed a trend towards clinical worsening, albeit not statistically significant (*p* = 0.137) (Figure 1). Therefore, the assessment of ExPH may predict a trend of increasing CV risk score over time.

The N-terminal prohormone of brain natriuretic peptide (NT-proBNP) exhibited a statistically significant difference at the end of the two-years follow-up between patients with low versus those with intermediate/high PH probability during exercise, which was higher in the group with intermediate/high PH probability (44 vs. 10 pg/mL in the intermediate/high probability vs. low probability of ExPH respectively, *p* = 0.033) (Figure 2). Even comparing individuals with and without ExPH at CPET, NT-proBNP values were significantly higher at the end of the two-years follow-up in patients with ExPH (62.5 vs. 24 pg/mL, *p* = 0.048) (Figure 3).

## 4. Discussion

We evaluated the existence of an association between ExPH assessed by CPET and ESE and the worsening of CV risk as measured by the two-year score in HIV patients with low or intermediate probability of PH at resting echocardiogram. We found that the presence of intermediate/high probability of ExPH at baseline during ESE was associated with a statistically nonsignificant trend of increased CV risk score after two years of follow-up compared with low probability of ExPH, suggesting a worse prognosis in this group of patients with ExPH. We found no statistically significant difference or a trend of difference in CV risk score in patients with ExPH during CPET at baseline compared with patients without ExPH. The lack of statistical significance of the difference in the CV risk score currently adopted by the ESC/ERS Guidelines in the two groups of patients might be due to the brevity of follow-up. A longer observation period would probably lead to a greater separation of risk score trends over time between the two patient groups, achieving statistical significance. Overall, our preliminary results support the hypothesis and the continuation of the study and lay the foundation for considering the utility of ESE in predicting the CV risk of patients with HIV.

In the present study, we excluded patients who had a high probability of pulmonary hypertension at rest at baseline (time 0, i.e., patient enrollment), because these patients likely have more advanced pulmonary artery abnormalities. Measuring ExPH in these patients would not add value for their clinical and risk assessment. Our intention was to detect patients with early vascular lesions, i.e., those who do not have pulmonary hypertension at rest (in probabilistic terms, those who have a low or intermediate probability of PH), but who develop ExPH as a consequence of the presence of early onset vascular disease. This could lead to a reduction in the pulmonary vascular reserve due to a greater rigidity of the pulmonary vessels and, therefore, a lower vasodilatory response capacity to physical exercise.

The exact mechanisms of ExPH in HIV patients are unknown. However, endothelial dysfunction triggered by exercise as a consequence of reduced pulmonary vascular reserve would represent the most accredited mechanism. In comparison, in healthy individuals, sPAP or tricuspid regurgitation velocity does not exceed 40 mm Hg and 2.7 m/s with exercise, respectively, except for athletes [19]. In patients at high risk of PH (e.g., patients with connective tissue disease or HIV infection), we and others have well documented the development of ExPH, independent of high cardiac output or retrograde transmission of high filling pressures due to left heart disease [17,18,20,21,22,23,24,25,26,27,28]. In the present study, the echocardiographic variables of left ventricular function were normal at rest, and no data of left ventricular systolic or diastolic dysfunction were observed. Thus, we consider it unlikely that our patients had pulmonary venous hypertension at rest. At peak exercise, we did not observe differences between the two groups in terms of cardiac output, and this allows us to rule out the presence of pulmonary venous hyperflow as a cause of ExPH.

A biomarker of heart failure, NT-proBNP is released from the heart when the heart walls are strained or there is an overload of pressure on the heart. In left heart disease, impaired exercise capacity [29] and poor prognosis [30,31,32,33] are associated with elevated NT-proBNP levels, while their correlate is associated with right ventricle dysfunction and increased mortality in patients with idiopatic PAH or with CTEPH (chronic thromboembolic PAH) [34,35,36,37]. In the present study, we found that ExPH at baseline, assessed by both ESE or CPET, correlated statistically significantly with the increase in NT-proBNP at the end of the two-year follow-up. At the same time, the population with ExPH at baseline assessed at both ESE and CPET, which had increased NT-proBNP levels at the end of the two-year follow-up, also showed echocardiographic parameters indicative of right ventricular systolic and diastolic dysfunction, as well as increased total pulmonary vascular resistance. Thus, the present study supports the hypothesis that ExPH at baseline is a predictor of right ventricle dysfunction and worsening pulmonary hemodynamics in patients with HIV.

The absence of a diagnosis of ExPH by cardiac catheterization, the small sample size and the short follow-up are major limitations of the present study. We did not find a higher CV risk score in the ExPH group, but only a worsening trend, probably due to the short follow-up.

## 5. Conclusions

The assessment of ExPH may predict a trend of increasing CV risk score over time, which, if confirmed after longer follow-up, could contribute to better risk stratification in HIV patients. Assessment of ExPH by CPET or ESE may contribute to better stratification of patients with HIV, who tend to develop right ventricular dysfunction and worsening in pulmonary hemodynamics, and, thus, to the planning of more adequate follow-up.

## Figures and Tables

**Figure 1 jcm-11-07349-f001:**
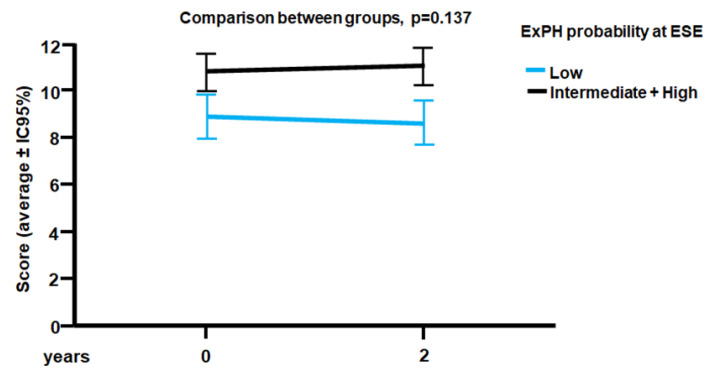
ESC/ERS CV risk score at baseline and at the end of the two-years follow-up, according to ExPH probability at ESE. CV: cardiovascular; ERS: European Respiratory Society; ESC: European Society of Cardiology; ExPH: exercise-induced pulmonary hypertension; ESE: Stress echocardiography.

**Figure 2 jcm-11-07349-f002:**
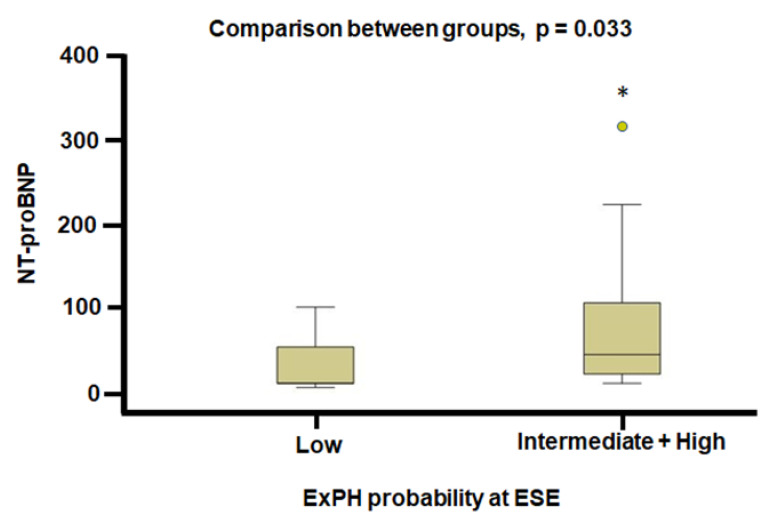
NT-proBNP values at the end of the two-years follow-up in patients with low and intermediate/high probability of PH. NT-proBNP: N-terminal prohormone of brain natriuretic peptide; ExPH: exercise-induced pulmonary hypertension; ESE: Stress echocardiography, * *p* < 0.05.

**Figure 3 jcm-11-07349-f003:**
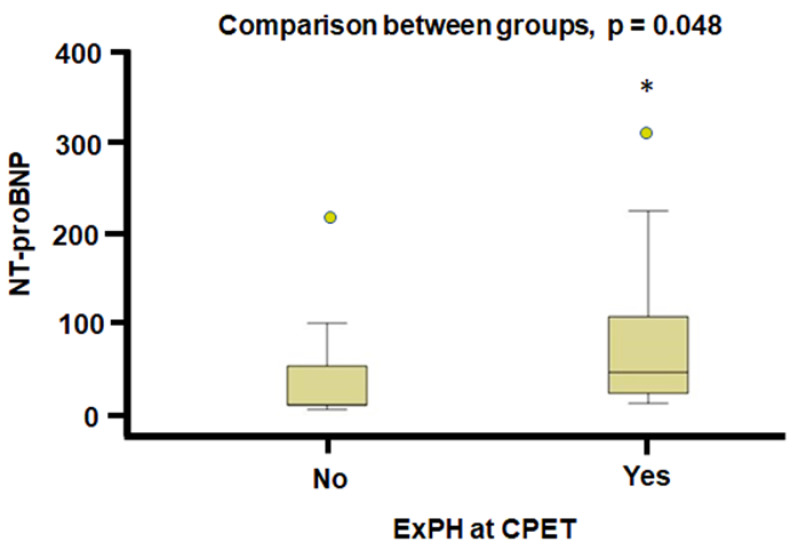
NT-proBNP values at the end of the two-years follow-up in patients with and without ExPH at CPET. CPET: cardiopulmonary stress test, ExPH: exercise-induced pulmonary hypertension, NT-proBNP: N-terminal prohormone of brain natriuretic peptide, * *p* < 0.05.

**Table 1 jcm-11-07349-t001:** Comparison between EXPH at CPET and echocardiographic parameters at rest.

Parameter	ExPH No	ExPH Yes	*p*-Value
**LVEDD (mm)**	46 (43–52)	43 (40–46)	0.123
**LVESD (mm)**	29 (25–35)	24 (20–34)	0.146
**LVEDV (mL)**	104 (82–125)	91 (76–117)	0.411
**iLVEDV (mL/m^2^)**	36 (30–43)	33 (25–39)	0.232
**LVESV (mL)**	40 (28–46)	35 (21–47)	0.537
**iLVESV (mL/cm^2^)**	13 (10–16)	12 (10–15)	0.411
**LVmass (g)**	160 (131–194)	151 (104–178)	0.243
**iLVmass (g/m^2^)**	77 (57–93)	63 (40–86)	0.208
**LAD (mm)**	48 (42–52)	49 (39–52)	0.888
**LAV (mL)**	35 (22–50)	32 (22–38)	0.404
**iLAV (mL/m^2^)**	12 (7–23)	13 (5–30)	0.517
**LVEF (%)**	65 (59–70)	61 (60–70)	0.908
**E wave (cm/s)**	70 (56–80)	77 (61–87)	0.154
**A wave (cm/s)**	70 (57–82)	79 (63–90)	0.243
**Septal E wave (cm/s)**	10 (7–12)	10 (5–10)	0.194
**E/A**	0.9 (0.7–1.4)	0.9 (0.7–1.4)	0.990
**E/e’**	6 (5–9)	8.1 (6.8–11)	**0.039**
**iRVESA (cm^2^/m^2^)**	4.1 (3.2–5.1)	5.3 (4.5–5.9)	0.070
**iRVEDA (cm^2^/m^2^)**	6 (5–8)	7 (6–9)	0.425
**iRVESV (mL/m^2^)**	6 (3–8)	5 (3–7)	0.877
**iRVEDV (mL/m^2^)**	15 (10–18)	13 (9–16)	0.322
**RD1 (mm)**	37 (33–40)	34 (32–42)	0.441
**RD2 (mm)**	32 (28–35)	28 (24–35)	0.129
**RD3 (mm)**	19 (15–23)	15 (11–22)	0.169
**RVOT prox (mm)**	32 (28–38)	34 (30–35)	0.888
**RVOT dist (mm)**	34 (30–39)	33 (30–37)	0.537
**Eccentricity index**	0.9 (0.9–1)	0.9 (0.8–1)	0.501
**Right ventricle/left ventricle basal diameter ratio**	0.8 (0.7–0.9)	0.8 (0.7–0.9)	0.443
**TAPSE (mm)**	22 (19–24)	16.5 (15.5–17)	**<0.001**
**FAC (%)**	51 (46–54)	43 (39–49)	**0.010**
**E_RV_/ARV**	1.4 (0.8–1.7)	0.7 (0.6–0.9)	**0.001**
**E_RV_/e′**	3.2 (2.1–5.2)	2.7 (2.4–4.3)	0.721
**S_RV_ vel**	12 (10–14)	12 (9–14)	0.802
**S_RV_ VTI**	2.6 (2.1–3.4)	1.9 (1.7–2.3)	**0.047**
**sPAP(mmHg)**	21 (19–27)	35 (30–41)	**<0.001**
**mPAP (mmHg)**	15 (13–18)	20 (19–25)	**0.004**
**TR velocity (m/s)**	2 (1.8–2.3)	2.8 (2.1–3)	**0.006**
**Right ventricular out flow doppler acceleration (t/s)**	110 (81–130)	120 (116–147)	0.108
**VCI diameter (mm)**	16 (13–17)	16 (15–17)	0.765
**Right atrial area (cm^2^)**	14 (12–16)	21 (20–25)	**<0.001**
**iRAV (mL/m^2^)**	5.3 (4.1–6.9)	8.1 (6.2–11.1)	**0.004**
**VTI rvot**	15 (13–16)	13 (12–14)	0.133
**VRT/VTI rvot**	0.1 (0.1–0.2)	0.3 (0.2–0.3)	**<0.001**
**TAPSE/PAPs**	1 (0.8–1.3)	0.9 (0.4–1.4)	0.279
**Concentric_remodeling**			0.258
no	20	5	
yes	13	7	
**Normal_geometry**			0.286
no	19	9	
yes	14	3	
**Concentric_hypertophy**			0.721
no	29	11	
yes	4	1	
**Eccentric_hypertrophy**			0.787
no	31	11	
yes	2	1	
**MR**			0.366
no	17	8	
yes	16	4	
**AO**			0.552
no	28	11	
yes	5	1	

Statistics: median (IQR) or absolute frequency (sd). Abbreviations: LVEDD, left ventricle end-diastolic diameter; LVESD, left ventricle end-systolic diameter; LVEDV, left ventricle end-diastolic volume; LVESV, left ventricle end-sistolic volume; iLVEDV, indexed LVEDV; iLVESV, indexed LVESV; LAD, left atrial diameter; LAV, left atrial volume; iLAV, indexed LAV; LVEF, left ventricle ejection fraction; iRVESA, indexed right ventricle end-sistolic area; iRVEDA, indexed right ventricle end-diastolic area; iRVESV, indexed right ventricle end-sistolic volume; iRVEDV, indexed right ventricleend-diastolic volume; RD, right diameter; RVOT, right ventricle outflow; TAPSE, tricuspid annular plane excursion; FAC, fractional area change; VTI, velocity time integral; sPAP, sistolic pulmonary arterial pressure; mPAP, mean PAP; TR, tricuspid regurgitation; IVC, inferior vena cava; iRAV, indexed right atrial volume; TRV, tricuspid regurgitation velocity; PVRI, Pulmonary Vascular Resistance Index. ExPH, exercise-induced pulmonary. hypertension; CPET, cardiopulmonary exercise test.

**Table 2 jcm-11-07349-t002:** Comparison between ExPH probability at ESE and echocardiographic parameters at rest.

Parameter	Low	Intermediate + High	*p*-Value
**LVEDD (mm)**	46 (42–52)	45 (41–49)	0.608
**LVESD (mm)**	28 (23–33)	29 (21–36)	0.891
**LVEDV (mL)**	106 (88–125)	96 (77–123)	0.426
**iLVEDV (mL/m^2^)**	36 (30–41)	33 (27–41)	0.516
**LVESV (mL)**	41 (25–46)	35 (26–47)	0.480
**iLVESV (mL/cm^2^)**	14 (10–16)	12 (10–15)	0.426
**LVmass (g)**	162 (136–194)	150 (118–181)	0.195
**iLVmass (g/m^2^)**	77 (57–94)	66 (44–91)	0.333
**LAD (mm)**	51 (42–55)	46 (41–51)	0.165
**LAV (mL)**	44 (27–50)	31 (20–39)	0.065
**iLAV (mL/m^2^)**	11 (8–15)	13 (8–17)	0.523
**LVEF (%)**	65 (60–70)	62 (59–70)	0.864
**E wave (cm/s)**	72 (56–85)	71 (59–82)	0.955
**A wave (cm/s)**	73 (58–84)	69 (57–86)	0.608
**Septal E wave (cm/s)**	10 (7–12)	10 (7–12)	0.698
**E/A**	0.9 (0.7–1.4)	0.9 (0.7–1.4)	0.622
**E/e′**	7 (5–10)	7 (6–9)	0.855
**iRVESA (cm^2^/m^2^)**	4 (3–5)	5 (3–6)	0.284
**iRVEDA (cm^2^/m^2^)**	6 (5–8)	7 (5–9)	0.945
**iRVESV (mL/m^2^)**	5 (3–12)	5 (1–10)	0.855
**iRVEDV (mL/m^2^)**	15 (12–18)	13 (9–16)	0.380
**RD1 (mm)**	37 (33–40)	34 (32–40)	0.439
**RD2 (mm)**	33 (29–35)	30 (26–34)	0.148
**RD3 (mm)**	18 (17–23)	18 (13–24)	0.508
**RVOT prox (mm)**	33 (28–38)	34 (29–35)	0.909
**RVOT dist (mm)**	34 (30–38)	33 (28–38)	0.546
**Eccentricity index**	0.9 (0.9–1.1)	0.9 (0.9–1)	0.156
**Right ventricle/left ventricle basal diameter ratio**	0.8 (0.7–0.9)	0.8 (0.8–0.9)	0.558
**TAPSE (mm)**	21 (18–24)	20 (17–23)	0.186
**FAC (%)**	52 (45–55)	49 (43–51)	0.092
**E_RV_/ARV**	1.4 (0.8–1.7)	0.9 (0.7–1.5)	0.298
**E_RV_/e′**	3.1 (2.3–6)	2.8 (2.6–3.2)	0.334
**S_RV_ vel**	12 (9–14)	13 (10–16)	0.338
**S_RV_ VTI**	2.6 (1.9–3.1)	2.2 (1.9–3.2)	0.866
**sPAP(mmHg)**	20 (18–27)	29 (23–35)	**0.005**
**mPAP (mmHg)**	14 (13–18)	18 (16–22)	**0.019**
**TR velocity (m/s)**	1.9 (1.6–2.3)	2.3 (1.9–2.8)	**0.009**
**Right ventricular out flow doppler acceleration (t/)**	106 (77–130)	119 (101–139)	0.183
**VCI diameter (mm)**	15 (12–16)	16 (15–17)	0.073
**Right atrial area (cm^2^)**	13.5 (11.1–16)	19 (14.6–22.5)	**0.010**
**iRAV (mL/m^2^)**	5 (4–7)	6 (5–9)	0.168
**VTI rvot**	15 (13.2–16.6)	13.6 (12.2–15.3)	0.062
**VRT/VTI rvot**	0.1 (0.1–0.1)	0.2 (0.2–0.3)	**0.004**
**TAPSE/PAPs**	1.1 (0.8–1.5)	0.9 (0.7–1.3)	0.230
**Concentric_remodeling**			0.161
no	14	11	
yes	7	13	
**Normal_geometry**			0.967
no	13	15	
yes	8	9	
**Concentric_hypertophy**			0.133
no	17	23	
yes	4	1	
**Eccentric_hypertrophy**			0.472
no	19	23	
yes	2	1	
**MR**			0.316
no	10	15	
yes	11	9	
**AO**			0.860
no	18	21	
yes	3	3	

Statistics: median (IQR) or absolute frequency (sd). Abbreviations: LVEDD, left ventricle end-diastolic diameter; LVESD, left ventricle end-systolic diameter; LVEDV, left ventricle end-diastolic volume; LVESV, left ventricle end-sistolic volume; iLVEDV, indexed LVEDV; iLVESV, indexed LVESV; LAD, left atrial diameter; LAV, left atrial volume; iLAV, indexed LAV; LVEF, left ventricle ejection fraction; iRVESA, indexed right ventricle end-sistolic area; iRVEDA, indexed right ventricle end-diastolic area; iRVESV, indexed right ventricle end-sistolic volume; iRVEDV, indexed right ventricleend-diastolic volume; RD, right diameter; RVOT, right ventricle outflow; TAPSE, tricuspid annular plane excursion; FAC, fractional area change; VTI, velocity time integral; sPAP, sistolic pulmonary arterial pressure; mPAP, mean PAP; TR, tricuspid regurgitation; IVC, inferior vena cava; iRAV, indexed right atrial volume; TRV, tricuspid regurgitation velocity; PVRI, Pulmonary Vascular Resistance Index. ExPH, exercise-induced pulmonary hypertension; ESE, stress echocardiography.

## Data Availability

The data are deposited in the electronic database (excel file) of our institution and a copy is saved in the electronic database (SPSS file) of our statistician.

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
