# Peer review of "Impact of Exercise-Induced Pulmonary Hypertension on Right Ventricular Function and on Worsening of Cardiovascular Risk in HIV Patients"

_jcm, 2022, doi:10.3390/jcm11247349_

Round 1

Reviewer 1 Report

Brief summary: The authors present for consideration their report of their observational, prospective, cohort study of 52 patients with HIV at risk for the development of pulmonary hypertension.  They propose to use their data set including baseline to examine whether the presence of exercise related pulmonary hypertension may serve to predict the development of increased cardiovascular risk in the future. 

General concept comments
The question is presented has the potential to be interesting and important.  While evaluation of exercise-induced pulmonary hypertension is not itself novel, looking at it specifically in the context of patients with HIV does have some novelty.  Because of their detailed and prospective data on both echocardiographic and metabolic parameters, the authors appear to have sufficient resources to make an interesting contribution to this area of study.  Unfortunately, problems with the study design and in particular analysis plan appear to prevent the authors from being able to make any meaningful analyses or conclusions.  Problems with the presentation of the data also make it difficult to interpret the findings they do have.

Major weaknesses summary:

The authors state (line 74) that “ExPH may indicate an early stage of pulmonary vascular disease and therefore correlate with reduced survival, similar to individuals with resting PAH.”  They then state that in this manuscript they will discuss the results of an integrated diagnostic evaluation that would permit them to recognize patients with HIV early in the course of their pulmonary vascular disease.  These are compelling statements.  Unfortunately, it appears that they made the decision to exclude those patients who, despite the absence of pulmonary hypertension at rest at the baseline assessment, went on to develop resting pulmonary hypertension.  Those people are precisely the ones who would be most useful to include in the analysis as their presence suggests that ExPH may indeed represent an earlier clinical entity.  Excluding these people would be expected to contribute to type II error, biasing the results of the study towards the null hypothesis.  In the absence of significant findings, the authors go on to make conjectures about trends that are not strongly supported by the evidence.  This combination of flaws – possible Type II error and unsupported conclusions, leads me to recommend against accepting manuscript in its current form.

Specific comments:

The following specific comments are intended to serve as examples and instances of them major weaknesses described above:

Line 197: “6 patients were excluded at follow-up since at high probability of PH”

While the sentence as constructed is ambiguous, it appears that the authors excluded patients who did not have high probability of PH at baseline, but later demonstrated high probability of PH at follow-up.  This is a puzzling decision, and the rationale is not clearly explained.  Patient found to have PH at follow-up are exactly the patients where one would hope to query regarding the utility of isolated ExPH as a predictor.  If these patients had ExPH at baseline, and later developed PH, this help the authors being to evaluate which particular parameters on the baseline testing were abnormal, and thus serve to generate hypotheses for future study.  If the only patients who went on to develop PH at follow-up did not demonstrate ExPH at baseline, this would provide potentially important data arguing against the utility of exercise testing for PH as a predictor of future PH.  While perhaps a disappointing result for clinicians enthusiastic about exercise testing in PH, it would nonetheless be a valuable important finding, permitting better allocation of resources and patient time. Instead, by excluding these patients, we are left without the most important data. 

If instead I misunderstood this statement, then I would ask the authors to clarify it for the reader. 

Line 208: “However, RV systolic function was significantly reduced in patients 208 with ExPH, who exhibited worse values of TAPSE and FAC” Table 1.

In this line and all of the discussion of the results that follows, it is entirely unclear from the tables and text whether the numerical values presented represented the values from the resting echo or from the exercise echo.  I would have expected that both the resting and exercise values would be presented, but without doing that it is impossible to interpret the meaning of any of these values or there analyses.

In addition, the title of table 2, “Table 2. Comparison between ExPH probability at ESE and echocardiographic parameters.” introduces further ambiguity for the same reason.  For example, patients with intermediate and high probability of pH would be expected to have a higher TR velocity, systolic pulmonary artery pressure, and estimated right atrial pressure.  Performing a statistical test on the key variables that contribute to the definition of the group does not seem to have much probative value.

Line 254: “whereas at the end of the follow-up individuals with intermediate/high PH probability showed a trend towards clinical worsening, albeit not statistically significant (p=0.137) (Figure 1). Thus, the evaluation of ExPH may predict a future increase of CV risk score over time.

While the first sentence gives appropriate caveats, the second is indicative of an unsupported conclusion based on the evidence as presented.  This type of interpretation is carried through the remainder of the manuscript and essentially forms the basis for the final conclusion, which is not warranted based on the data in the manuscript.

Line 253-271

The interpretation of the results of NT proBNP testing are limited by the presentation of the data.  It is unclear whether the authors are presenting actual values for NT proBNP, or the change in NT proBNP over time in these patients.  Merely stating that patients with exercise-induced pulmonary hypertension have a slightly higher NT proBNP than patients without exercise-induced pulmonary hypertension is not particularly important or unexpected.  Especially since the mean NT proBNP in the exercise pH group appears to be only 44 vs 10 pc/ml, still entirely within normal limits.  If instead it remained normal, but increased to a greater degree in the exercise-induced pH group, that could represent something useful prognostically above and beyond the utility of NT proBNP, which is already well established.

Lines 311-313

The authors state that echocardiographic variables of left ventricular function were normal at rest and during exercise, but appear to fail to report the actual echocardiographic variables with exercise in either the table or the online supplement.  They then use that on reported data to make the statement that they consider it unlikely that the patient’s with exercise-induced pH had pulmonary venous hypertension.  This disregards the data that is presented within Table 1 which shows that the E/e’ was greater in the ExPH group compared with the no ExPH group, suggesting greater left sided filling pressures in that group.  It would be useful to present the exercise value for E/e’ which has been shown to correlate with PCWP at exercise as well.

Recommendation: While not acceptable in its current form, I would be glad to review the paper if revised and submitted after the authors 1) repeat their analyses including the patients who later developed pulmonary hypertension at follow-up but did not have resting PH at baseline, 2) examining the ability of the baseline exercise test to predict future changes (delta) as opposed to future static values. 3) included post-exercise echocardiographic variables in another table, 4) clarified the writing where ambiguous and 3) reworked their conclusion statements limiting themselves to only statements supported by the results of their experiments.  Fortunately, due to what appears to be an excellent plan for prospective data collection, I am optimistic that the authors have within their possession all of the necessary data to make these improvements to their manuscript.

Reviewer 2 Report

PH = mPAP>20mmHg

Precapillary PH = mPAP>20mmHg, PCWP<15, PVR >2 WU

ex PH = mPAP/CO slope between rest and exercise >3mmHg/L/min (exercise PH is not the increase of sPAP>20mmHg in exercise)

We need RHC.

Some of the measured echo values such as LAVi, RVESV seem to be not righ

Round 2

Reviewer 2 Report

there are no robust data on the prognostic validation of exercise induced PH

Many controversies exist

The accuracy of exercise schocardiography is not well established